# Evolution of the Ergot Alkaloid Biosynthetic Gene Cluster Results in Divergent Mycotoxin Profiles in *Claviceps purpurea* Sclerotia

**DOI:** 10.3390/toxins13120861

**Published:** 2021-12-02

**Authors:** Carmen Hicks, Thomas E. Witte, Amanda Sproule, Tiah Lee, Parivash Shoukouhi, Zlatko Popovic, Jim G. Menzies, Christopher N. Boddy, Miao Liu, David P. Overy

**Affiliations:** 1Ottawa Research & Development Centre, Agriculture & Agri-Food Canada, 960 Carling Ave., Ottawa, ON K1A 0C6, Canada; carmen.hicks@agr.gc.ca (C.H.); thomas.witte@agr.gc.ca (T.E.W.); amanda.sproule@agr.gc.ca (A.S.); tlee076@uottawa.ca (T.L.); parivash.shoukouhi@agr.gc.ca (P.S.); miaomindy.liu@agr.gc.ca (M.L.); 2Morden Research and Development Centre, Agriculture & Agri-Food Canada, 101 Route 100, Morden, MB R6M 1Y5, Canada; zlatko.popovic@agr.gc.ca (Z.P.); jim.menzies@agr.gc.ca (J.G.M.); 3Department of Chemistry & Biomolecular Sciences, University of Ottawa, Ottawa, ON K1N 6N5, Canada; chrisotpher.boddy@uottawa.ca

**Keywords:** *Claviceps purpurea*, fungal plant pathogen, biosynthetic gene cluster, ergot alkaloids, mycotoxins, untargeted metabolomics, mass spectrometry, sclerotia

## Abstract

Research into ergot alkaloid production in major cereal cash crops is crucial for furthering our understanding of the potential toxicological impacts of *Claviceps purpurea* upon Canadian agriculture and to ensure consumer safety. An untargeted metabolomics approach profiling extracts of *C. purpurea* sclerotia from four different grain crops separated the *C. purpurea* strains into two distinct metabolomic classes based on ergot alkaloid content. Variances in *C. purpurea* alkaloid profiles were correlated to genetic differences within the *lpsA* gene of the ergot alkaloid biosynthetic gene cluster from previously published genomes and from newly sequenced, long-read genome assemblies of Canadian strains. Based on gene cluster composition and unique polymorphisms, we hypothesize that the alkaloid content of *C. purpurea* sclerotia is currently undergoing adaptation. The patterns of *lpsA* gene diversity described in this small subset of Canadian strains provides a remarkable framework for understanding accelerated evolution of ergot alkaloid production in *Claviceps purpurea*.

## 1. Introduction

Ergot fungi of the genus *Claviceps* are phytopathogenic ascomycetes with the ability to parasitize over 400 monocotyledonous plant species, most notably affecting a large number of grasses and common cereals [1,2]. *Claviceps* infections are characterized by the formation of recalcitrant resting structures known as sclerotia. Sclerotia form as fungal hyphae invade unfertilized ovules of host plants via stigmas and pollen tubes during anthesis, replacing developing host seeds with compact masses of hardened fungal mycelium [3,4]. While dormant in the sclerotial state, the fungus is protected, ensuring its survival though environmental extremes until appropriate weather conditions are presented [5]. Fruiting bodies (ascomata) then develop on the sclerotium that forcefully eject ascospores into the air, which are then dispersed by the wind to infect flowering host plants, thus continuing the life cycle.

Ergot infection of cereal crops directly impacts grain quality and yield and has become increasingly problematic across Canadian provinces over the past two decades [6,7]. Ergot-infested grain must have sclerotia removed prior to shipping, or grain shipments become downgraded or rejected at the point of sale, resulting in decreased returns to farmers [6]. However, yield reduction is seen as a secondary importance when compared to the toxic effects of accidental consumption by humans and other animals [8]. Ergot sclerotia contain a wide variety of ergot alkaloids that constitute 0.5–2% of the entire sclerotium mass [9]. Sharing structural similarities to neurotransmitters, ergot alkaloids interact antagonistically or agonistically to α-adrenergic, dopamine, serotonin 5-HT and other receptors, impairing both psychological and motor functions [10]. Preventing or minimizing the occurrence of ergot alkaloids in our food and animal feed is, therefore, of great importance.

The ergot alkaloid biosynthetic pathway is well characterized (Figure 1), and associated biosynthetic gene clusters were identified from the genomes of several *Claviceps* spp. [11]. The presence or absence of various genes in the gene cluster has a direct impact on the diversity of ergot alkaloids produced by a species. Ergot alkaloids are generally divided into three subclasses: simple clavines, ergoamides (simple lysergic acid amides) and the more complex peptide-containing ergopeptides; however, all share the same initial biosynthetic pathway steps involved in the tetracyclic ring formation [12,13]. Ergoamide and ergopeptide biosynthesis initiates with LpsB, a unique monomodular non-ribosomal peptide synthetase (NRPS) that recognizes D-lysergic acid as a substrate, activates it as the AMP-ester and loads it on the LpsB carrier protein. Unique among NRPS pathway, the two downstream NRPSs, LpsA or LpsC, compete for LpsB binding and the subsequent elaboration of lysergic acid [14]. LpsC is a monomodular NRPS responsible for the incorporation of a single amino acid that produce the ergoamides. LpsA is a trimodular NRPS that incorporates three amino acids units through the progressive elongation to form ergopeptams, which the mono-oxygenase EasH then converts into the ergopeptines via oxidation and spontaneous cyclization [15]. Two homologs of *lpsA* encoding LpsA1 and LpsA2 are present in the biosynthetic gene cluster. Sequence variations in the second adenylation domain (A-domain) of the two LpsA enzymes are responsible for incorporation of phenylalanine or leucine in the ergopeptams and ergopeptines [16]. Characterization of an *lpsA1* deletion mutant in *C. purpurea* P1 suggests that LpsA1 is responsible for formation of the phenylalanine containing ergotamine and ergocristine ergopeptines [17].

In Canada, guidelines are placed on grain sclerotia content allowances that range from 0.01 to 0.1% depending on the host crop [18]. Previous studies showed that ergot count and weight are not predictive of diagnostically relevant ergot alkaloid concentrations [19] as ergot alkaloid constituents and concentrations within sclerotia are highly variable and directly dependent upon the producing species. *Claviceps purpurea* is documented as producing a structurally diverse range of ergopeptides due to the adenylation site promiscuity of LpsA and to the tandem duplication and neofunctionalization of the *lpsA* gene; these were recently denoted *lpsA1* and *lpsA2* [11,17]. In addition to increased alkaloid content and variability, *C. purpurea* has the widest reported host range of any *Claviceps* species, making it a major contributor of ergot alkaloid food contamination across Canada [8,20]. Due to the wide associated host range, *C. purpurea* is a species complex that has been the focus of taxonomic evaluation. Multigene phylogenetic studies have confirmed cryptic speciation within the taxon, recorded as G1–7, wherein G1 represents *C. purpurea sensu stricto* [21,22]. The European Food Safety Agency (Panel on Contaminants in the Food Chain) recently called for increased monitoring and improved knowledge on the relation between *C. purpurea* sclerotia and ergot alkaloids of concern, in particular: ergometrine, ergotamine, ergosine, ergocristine, ergocryptine (α- and β-isomers), ergocornine and their corresponding -inine (S)-epimers [23]. Information on variability of *C. purpurea* ergot alkaloid content across crop hosts is, therefore, required if knowledgeable regulatory decisions are to be made regarding acceptable levels of ergot alkaloids for livestock and human consumption [24].

In this study, we used UPLC-HRMS and an untargeted metabolomics approach to profile the alkaloid content of *C. purpurea* sclerotia from four different grain crops. Consistent trends in sclerotia ergot alkaloid content among grain crops separated the *C. purpurea* strains into two distinct metabolomic classes. *Claviceps purpurea* alkaloid profiles were correlated to genetic differences of the ergot alkaloid biosynthetic gene cluster from previously published isolate genomes and from newly sequenced, long-read genome assemblies of Canadian strains. Based on the gene cluster composition and unique polymorphisms, we hypothesize that the alkaloid content of *C. purpurea* sclerotia is undergoing adaptation that contributes to the diversity of ergopeptides produced.

## 2. Results

### 2.1. Untargeted Metabolomics Profiling

An initial unsupervised Euclidean Ward hierarchical clustering was performed individually for each host crop to reveal the underlying structure within the metabolomics data. Hierarchical clustering of the binary mass feature data showed a dichotomy in metabolite profiles amongst the isolates, separating the sclerotium extracts into two putative “metabolomic classes”, which was consistent for all four host crops (Figure 2A, and Appendix A). Class 1 consisted of isolates LM04, LM207, LM232 and LM233, while Class 2 consisted of isolates LM30, LM60, LM469 and LM474. The results of the discriminant multivariate analyses supported the observed Class dichotomy in *C. purpurea* sclerotia. The top 30 ranking mass features with the greatest importance in the random forest model classification (based on both mean decrease accuracy and mean decrease Gini) were found to be highly conserved between host crops and demonstrated large fold changes between classes (Figure 2B, and Appendix A). From the orthogonal partial least squares discriminant analysis (OPLS–DA) models generated for each host crop, the resulting S-plots separated the mass features associated with latent variable class separation (black dots in Figure 2C, and Appendix A), where positive values on the Y-axis represent mass features with greater association to Class 1, and negative values are mass features with greater association to the Class 2 isolates. When overlaid on the OPLS-DA S-plot, the top 30 mass features derived from the random forest analysis were found to occur at opposite ends of the Y-axis (red and blue dots in Figure 2C), indicating a high correlation of the selected mass features with the formation of the observed classes in both discriminant multivariate analyses.

An untargeted consensus binary heatmap representing mass feature frequency of occurrence in sclerotia produced on the four host crops was constructed to further explore the overall mass feature diversity amongst the strains and between classes (Figure 2D). When hierarchical cluster analysis was applied, the resulting class dichotomy was consistent across individual plant crop hosts. Clear groupings of mass features were found to occur at high frequencies (on all four hosts) and were consistent between *C. purpurea* isolate classes. A larger section of mass features was found to be infrequently observed (occurring on a single host by a specific isolate) and likely resulted from inherent differences in crop host- and/or *C. purpurea* isolate-specific metabolism. Most notable, however, were the consistent production of class specific mass features, having a high frequency of occurrence on all four hosts but only occurring in sclerotia from either Class 1 or Class 2 *C. purpurea* isolates (highlighted in Figure 2D). These mass features were the same mass features designated as being responsible for class separation in the discriminant multivariate analyses derived from the non-binary mass feature data. Further investigation (see below) confirmed that these class-specific mass features represented metabolites derived from ergot alkaloid biosynthesis (Appendix A).

### 2.2. Ergot Alkaloid Profiles Drive the Separation of Claviceps purpurea Isolate Classes

A molecular network was constructed to visualize ergot alkaloid/*C. purpurea* class associations using representative strains of both Class 1 and Class 2 sclerotium extracts from rye (Figure 3). Ergocristine/ergocristinine and ergotamine were the major ergopeptines produced and unique to LM60 (Class 2) sclerotia and ergovaline/ergovalinine, ergocornine/ergocorninine, ergocornam and ergocryptam were the major ergopeptines and ergopeptams produced and unique to LM04 (Class 1) sclerotia. Both ergosine and ergocryptine were also major metabolites produced by LM04, with minor production observed from LM60 sclerotium extracts from rye.

Clear and consistent trends in ergot alkaloid production were also evident in the frequency consensus heatmap (Figure 4). When produced, all of the ergopeptine- and ergopeptam-associated mass features were typically detected in sclerotium extracts from all four host crops surveyed. Trends observed in ergopeptine and ergopeptam production between *C. purpurea* classes were consistent with the trends observed from the molecular network. Ergotamine and ergocristine were the predominant ergopeptines having most abundant peak areas in Class 2 sclerotia and ergocryptine and ergocornine were the most abundant ergopeptines in Class 1 sclerotia. Both ergocornine (with the exception of LM30 and LM60) and ergocryptine and were detected in both Class 1 and 2 sclerotia from all four hosts; however, Class 2 sclerotia from all four hosts contained a significantly greater abundance of ergocornine and ergocyptine, as only very minor peak intensities were observed in Class 1 sclerotium extracts (LM469, LM474). Ergosine was observed in both Class 1 and Class 2 sclerotia, with a greater abundance associated with Class 1 isolates on all four crop hosts. Finally, ergoamide mass features associated with ergometrine were also detected in all Class 2 sclerotia and only two strains of the Class 1 sclerotia, with only minor production observed from all strains on all four host crops. The two Class 1 strains that did not produce ergometrine (LM04 and LM232, Figure 4) were reported to have an internal stop codon within the *lpsC* gene, the NRPS responsible for biosynthesis of ergoamides such as ergometrine ([25]—this issue).

Major ergopeptine signals from the Class 2 group were almost universally not detected in Class 1 isolate extracts, whereas the ergopeptines that were characteristic of Class 1 isolate extracts were consistently detected in the Class 2 extracts, albeit at very low intensities. When the molecular structures of the ergopeptines and ergopeptams associated with the Class dichotomy are compared, a clear trend is observed. Ergopeptines and ergopeptams in Class 1 lack phenylalanine incorporation at the second amino acid position (aa2) of the tripeptide moiety.

### 2.3. Ergot Alkaloid Gene Clusters Show Increased Rates of Mutation Associated with LpsA Gene Modules

A comparison of the ergot alkaloid biosynthetic gene clusters from LM04 and LM60, selected as representative isolates from Classes 1 and 2, respectively, as well as *C. purpurea* isolate 20.1 [26] and LM72 ([25]—this issue), revealed a higher degree of polymorphism and repetitive DNA elements surrounding the *lpsA* genes (Figure 5A). Higher proportions of inter-strain polymorphisms were localized to specific *lpsA* modules, showing as much as 20% polymorphic sites (Figure 5A, red line graphs), well above the 1.5% ‘background’ polymorphic frequency calculated by comparison of two other NRPS genes, *lpsB* and *lpsC*, present in the cluster. Patterns of site divergence between *lpsA* genes correlated with the metabolomics data in two ways. First, we detected a frameshift mutation caused by an indel in the sequence encoding the second condensation domain of LM04 *lpsA2* (Figure 5A) that prematurely terminates the encoded NRPS. As such, only LpsA2 is not expected to be expressed as a functional full length NRPS in the Class 1 strain LM04. Second, the sequence encoding the second adenylation, carrier protein and condensation domains of *lpsA1* in LM04 had approximately 18% polymorphic sites compared to the Class 2 strain LM60 as well as LM72 and *C. purpurea* 20.1. These polymorphisms included missense mutations that change the A-domain’s ‘specificity code’, which is predictive of the domain’s amino acid specificity during NRPS biosynthesis [27] (Appendix A).

Sequence analysis of the second A-domains from LM04 *lpsA1* shows it encodes residues that most closely match the leucine ‘specificity code’ identified by Stachelhaus et al. [27,28]. While substrate specificity predictions of A-domains in bacterial NRPS systems are fairly robust, they are known to be less so in fungal NRPS systems. Fungal A-domain predictions are often limited to gross physicochemical properties [28] and, in some cases, they fail entirely [29]. Together, these observations provide a genetic framework for understanding the metabolomic patterns detected in strains from Class 1 and Class 2. The Class 1 strain LM04 has only one functional copy of *lpsA*, *lpsA1*, which encodes substrate specificity in the second A-domain of LpsA1 for leucine and other branched hydrophobic amino acids rather than phenylalanine, leading to the formation of the observed Class 1 ergopeptines. This contrasts with the documented phenylalanine selectivity of LpsA2 in *C. purpurea* P1 [17].

To verify whether LpsA1 second A-domain sequences, from other strains grouped in Class 1, contained a similar leucine selective A-domain, we extracted the corresponding protein sequences from *lpsA1* and *lpsA2* from the long-read genomes compared in this study and previously published Illumina short-read genomes for the *C. purpurea* strains LM30, LM60, LM207, LM461, LM232, LM233, LM469 and LM474 [30]. The short-read genomes were fragmented at the *lpsA* regions; however, the second A-domain encoding sequences appeared intact. A phylogenetic tree of the aligned sequences (Figure 5B) supports the presence of two divergent *lpsA1* lineages correlating to Class 1 and Class 2 strains. This result supports the hypothesis that all Class 1 strains inherited a divergent second A-domain sequence with substrate specificity for leucine rather than phenylalanine. Additionally, we observed that the *lpsA2* second A-domain of the *C. purpurea* strain 20.1 was highly divergent from all of the other A-domains included in the analysis. It also lacked the predicted specificity for phenylalanine. However, the high levels of sequence divergence between *C. purpurea* 20.1 *lpsA2* and LM04 *lpsA1* suggests that it is unlikely that these two altered A-domain share a common origin (Figure 5C).

Following the prediction of a frameshift mutation in LM04 *lpsA2*, we looked for evidence of *lpsA2* disruption in the genomes of other Class 1 strains to explore the possibility that this lineage has consistent pseudogenization of *lpsA2*. The same C2-domain frameshift mutation was detected in one other genome, LM232 (Appendix A). Alignment of the fragments of putative *lpsA1* and *lpsA2* genes from LM207 and LM233 to the LM04 genes showed no obvious signs of disruption; however, we note that the Illumina assemblies from LM207 and LM233 are fragmented and would benefit from long-read sequencing to thoroughly investigate this possibility. Although LM72 was not selected for metabolomic studies, our genome analysis indicated there is a premature stop codon in *lpsA2* in this strain, disrupting a condensation domain (Figure 5A).

### 2.4. Transposons and Transposon Fossils in the LpsA1/LpsA2 Region

The intergenic regions between *lpsA1* and *lpsA2* contained numerous small (200–1000 bp) sequences that are duplicated in other regions in the respective genomes, including one that is inserted into the remnants of the truncated and pseudogenized *easH2* in LM04. This suggests these sequences are potentially transposable element fossils. Repetitive elements in the *C. purpurea* 20.1 and LM60 intergenic regions show some similarity to repetitive elements annotated as DNA transposons of the ‘MULE-MuDR’ type, whereas repetitive elements in the LM72 and LM04 genomes show similarity to the ‘TcMar-Tc1′ type DNA transposons annotated in the LM04 genome. The *easH2* copies, which were truncated compared to *EasH1* and predicted to be pseudogenized, had low sequence similarity, approximately 50% nucleotide identity between shared sequence lengths, and between 20.1/LM60 and LM72/LM04 isoforms. A putative transposon in LM72 between *easF* and *easG* was observed. As this study did not include metabolomic analysis of LM72 sclerotia, we did not pursue the potential effects of this putative transposon on ergot alkaloid production.

## 3. Discussion

In this study we classified Canadian *C. purpurea sensu stricto* strains into two broad categories based on ergot alkaloid profiles. We arbitrarily designated these groups as ‘Class 1’ and ‘Class 2’, and showed that the classes are dominated by either aliphatic hydrophobic residue-incorporating or phenylalanine-incorporating ergot alkaloids, respectively. Our identification of diversified intra-species ergot alkaloid profiles is generally consistent with previously published analyses of *C. purpurea* strains. Historically, ergopeptide production in *C. purpurea* has been quite diverse and varied between geographic locations and broader population groupings. A prominent early example of this trend compared strain ‘P1’, which was found to produce primarily ergotamine and ergocryptine, with strain ‘Ecc93’, a producer primarily of ergocristine [16]; however, we note the lineage of Ecc93 may need to be further investigated in light of newer *Claviceps* nomenclature [21]. *Claviceps purpurea* strain 20.1 appears to have a similar ergot alkaloid profile as strain P1, reportedly producing ergotamine, ergocryptine and ergometrine as major products [26]. The presence of divergent ergot alkaloid profiles was also previously described from broader genetic groupings of *C. purpurea sensu lato* (G1–G4); however, the profiles of *Claviceps purpurea sensu stricto* (G1) were more difficult to define [31,32]. For example, early work by Pažoutová et al. [31] described *C. arundinis* (G2a) sclerotia as consistent producers of ergocristine, ergosine and small amounts of ergocryptine; however, the G1 strains showed highly variable ergot alkaloid presences and relative abundances. More recent work by Negård et al. [2] showed that Norwegian *C. purpurea* G1 sclerotia contained a complex mixture of ergotamine, ergosine, ergocornine, ergocryptine and ergocristine as major constituents. In light of the present study, we suggest that the profiles generated by Pažoutová and Negård indicate that G1 strains have diverse LpsA A-domain specificities, which results in a broad spectrum of chemo-phenotypes. More recently, a detailed high resolution LCMS analysis of ergot alkaloid profiles from various *Claviceps* species was published, which included six strains of *C. purpurea* isolated from Canadian cereals [33]. The metabolomic classes we defined here broadly correlate with their results: four of the six *C. purpurea* strains’ profiles exhibited abundant ergotamine, ergocristine and ergostine mass feature peak areas (consistent with Class 2), and one strain showed abundant ergocornine, ergocryptine and ergovaline peak areas (consistent with Class 1). Taken together, our results contribute to a continuing understanding of the diversity detected in this species, and highlight the tension inherent in classifying strains based on chemotaxonomic vs. genetic criteria.

Our genetic analysis of the ergot alkaloid gene clusters from long-read genomes provides a framework to understand the underpinnings of chemical phenotype diversity within the *C. purpurea* G1 group. The ergot alkaloid biosynthetic gene clusters extracted from long-read assemblies of LM04 and LM60, chosen as representatives from the Class 1 and Class 2 metabolomic groupings, support a hypothesis that the class-specific ergot alkaloid profiles result from sequence diversity in the *easH*/*lpsA* tandem-duplicated region [16,17]. We predict that these mutations cause the shifting of substrate specificities of the *lpsA1* second A-domain to favor leucine and related aliphatic hydrophobic substrates such as isoleucine and valine, and additionally propose that *lpsA2* has been pseudogenized, at least in LM04, restricting ergotamine biosynthesis to interactions between LpsB and LpsA1. Given that the specific frameshift mutation described above is only detected in the genomes of LM04 and LM232, but not the other Class 1 strains LM233 or LM207, it is possible that the frameshift is not a causal feature for the metabolomic shift away from phenylalanine-incorporating ergot alkaloids across all Class 1 strains, but rather, could be a symptom of an overall relaxed selection for LpsA2 functionality due to some other mechanism. Importantly, this relaxed functionality is potentially not limited to Class 1 strains, but may also be present in Class 2 strains, for which both the *lpsA1* and *lpsA2* second A-domains are predicted to have phenylalanine specificity. The identification of a premature stop codon due to a single nucleotide polymorphism in LM72 *lpsA2* supports this hypothesis. Although LM72 has yet to associated with a given metabolomic class, we hypothesize it is in Class 2 based on A-domain selectivity predictions. The lack of detected phenylalanine-containing ergotamines in Class 1 metabolomes strongly suggests that the *lpsA2* gene is nonfunctional at least in those strains. Notably, this pattern departs from the known ergot alkaloid biosynthetic profiles of *C. purpurea* P1, in which researchers found that the *lpsA1* and *lpsA2* genes were both active, and the ergot alkaloid output of strains with *lpsA1* knocked out lost the ability to produce phenylalanine containing ergotamine but still produced leucine containing ergocryptine [17,34].

The ergot alkaloid chemical phenotype patterns observed in this study indicate that a representative portion of the Canadian populations of *C. purpurea* produce either exclusively short-chain alkyl group-bearing products (Leu, Ile, Val), or primarily phenylalanine-incorporating products with trace amounts of short-chain alkyl group-bearing products. At this time, the advantages of either phenotype on plant colonization, if any, are unknown, limiting the utility of speculation on the selective pressures shaping the evolution of *lpsA*. One important question that remains unanswered is: what are the origins of the mutations? A larger population of genomic and metabolomic data is likely needed to form a robust hypothesis; however, the presence of transposable elements with similarity to the MULE and TcMar families of DNA transposons detected in the intergenic space of the *lpsA* genes strongly suggests that the TE-mediated transposition or mutations associated with TE insertions are potential contributors to the *lpsA* diversity. Notably, the repetitive element-rich, highly polymorphic *lpsA1/lpsA2* intergenic spaces associated with either *C. purpurea* 20.1/LM04 or LM60/LM72 (Figure 5) are tightly linked with divergent *lpsA2* C0 domain sequences (Appendix A), thereby linking the presence of repetitive element insertions to nearby gene-coding sequence changes. Although the pattern of linked intergenic and *lpsA2* C0 domains is reflected in all genomes included in this study, the resulting groupings do not perfectly correlate with the assigned metabolomic classes of the strains. The domain-specific, highly variable proportions of *lpsA1/2* mutations detected when comparing strains (for example, see the comparison of *lpsA1* between LM04 and LM72, Figure 5) suggests that *lpsA* genes are likely undergoing recombinational shuffling [35,36]. Our analysis highlights the highly dynamic nature of the tandem-duplicated *easH*/*lpsA* region and the nature of ergot alkaloid NRPS module-specific evolution.

All ergot alkaloids contribute to and are responsible for a broad spectrum of biological activities as they can act as agonists, partial agonists and antagonists of multiple receptor sites, located throughout the body in various tissue types [37]. Therefore, the effects of ergot alkaloids on mammalian systems such as humans and livestock (i.e., cattle, sheep, horses and goats) are diverse, where, in the case of livestock, observed symptomatic responses to ergot alkaloid exposure can be highly variable and, therefore, historically difficult to diagnose [38]. Gangrenous ergotism occurs most frequently from acute ergot alkaloid exposure, caused by general blood vessel vasoconstriction and dysfunction resulting in tissue necrosis (dry gangrene) of the extremities such as the ear tips, tail, lower limbs and hooves [37,38,39]. In other instances, the consumption of subtle quantities of—and the prolonged exposure of livestock to—ergot alkaloids negatively impacts energy metabolism, feed efficiency and livestock productivity, as evidenced by decreases in food intake, live weight gain, circulating prolactin, reproductive performance, milk production and hyperthermia [37,38,39]. Pregnant and lactating animals are most susceptible to ergot alkaloid exposure due to increased risk of abortion and agalactia syndrome (the absence or failure to secrete milk); there is a direct correlation of ergopeptide exposure with a decrease in prolactin, growth hormone and luteinizing hormone secretion, and the inhibition of milk production [37]. Increasing our current understanding regarding the variability in production of ergot alkaloid content in *C. purpurea* sclerotia derived from different cereal crops used for food and feed production is, therefore, of continued relevance.

Ergot alkaloid profiles from *Claviceps* sclerotia consist of complex mixtures of minor stereoisomers, constitutional isomers and transient intermediate products, resulting in numerous peaks with the same *m*/*z* being detected, often at very low abundances when compared to the dominant *m*/*z* peak [31]. This was reflected in our analysis as numerous mass features were observed that shared duplicated isobaric *m*/*z* with ergotamine, ergocornine, ergocryptine and ergosine isomers, as well as isomers of other lactam and precursor molecules. However, many of these isobaric mass features were found to have random occurrences across strains and plant crop hosts, often occurring at a low abundance just above mass detection thresholds. The main ergoamides and ergopeptides of regulatory concern [23], namely ergometrine, ergotamine, ergosine, ergocristine, ergocryptine and ergocornine, when produced, were found to occur in *C. purpurea* sclerotia regardless of host crop. Of the *C. purpurea* strains examined, ergotamine and ergocristine were consistently the most abundant ergopeptides observed in Class 2 sclerotia, and ergocornine and ergocryptine were the most abundant in Class 1 sclerotia, from all four host crops tested. Based on the limited sample size, these four ergopeptides are likely to be of greatest concern in food and feed commodities derived from rye, triticale, durum wheat and ‘common’ wheat; however, a more extensive sampling of these crops using a larger breath of *C. purpurea s.s.* isolates will first be required to bring credence to this assumption.

## 4. Conclusions

The patterns of *lpsA* gene diversity described in this small subset of Canadian strains provides a remarkable framework for understanding the accelerated evolution of ergot alkaloid profiles. The results of this study provide insight into the variation in alkaloid content across Canadian *C. purpurea* isolates that could help guide future experiments in the exploration of secondary metabolite production. This research is crucial for providing developments in the understanding of the toxicological impacts of *Claviceps* species in order to improve consumer safety. *Claviceps purpurea* alkaloid content is an important topic of investigation because of its recognition as a major impactor of Canadian crops. The eight *C. purpurea* isolates that were analyzed within this study show clear differentiation of ergot alkaloid production. To impact the current regulatory mandate, the next steps would involve analyzing a larger sample size, with isolates across a broader range of hosts and host replicates as well as a broader geographic distribution to develop a better understanding of metabolomic class distribution. An additional recommendation that can be made based on the dichotomy in the observed ergot alkaloid production from *C. purpurea s.s.* isolates is that care should be observed when extrapolating taxonomic inferences based solely on ergot alkaloid profiles from sclerotia of unknown origin.

## 5. Materials and Methods

### 5.1. Strain Selection and Greenhouse Inoculation

Eight isolates of *C. purpurea* obtained from different Canadian provinces were selected from the Canadian Fungal Culture Collection (DAOMC) and the Liu lab research collection for in planta inoculation and untargeted metabolomic profiling (Appendix A). From each isolate, representative sequences of the *RPB2* and *TEF1*-*α* protein coding genes were obtained, concatenated, and used to construct a maximum parsimony analysis with various *Claviceps* spp.; all eight selected isolates were classified as *C. purpurea s.s.* (Appendix A). In planta inoculations were completed on four different crop species: rye (*Secale cereale* L.), the ‘common’ wheat cultivar AC Cadillac (*Triticum aestivum* L.), durum wheat (*Triticum durum* Desf.) and triticale (*x Triticosecale* Wittmack). *Claviceps purpurea* isolates were grown on potato dextrose broth to obtain spores and spore viability was checked by plating onto malt extract agar. Viable spores were inoculated at a concentration of 10,000 conidia per mL on 20 florets per spike with 3 replicates per host species. Five healthy spikelets on each side of the spike were selected and the primary and secondary florets were filled with the conidial suspension using a syringe and hypodermic needle [40]. The sclerotia were collected at crop maturity when the infection was successful. Greenhouse inoculation trials yielded sufficient *C. purpurea s.s.* sclerotia (n = 6/host, with the exception of a single isolate LM60, which only yielded three sclerotia when inoculated on durum wheat) from the four cereal host plants. Each harvested sclerotium were cut into 5 mg fragments, with one fragment subjected to DNA extraction and *easE* gene sequencing to confirm fungal identity and the second fragment reserved for alkaloid profiling.

### 5.2. Isolate Identity Confirmation

Genomic DNA was extracted from sclerotium fragments using a Macherey–Nagel NucleoMag 96 Trace Kit (Macherey–Nagel, Düren, Germany) following a modified protocol as described [22]. Polymerase chain reaction (PCR) amplification was performed on all extracted gDNA samples in 10 μL reactions. Final concentrations of 1× Titanium Taq buffer (with 3.5 mM MgCl_2_), 0.1 mM dNTPs, 0.08 µM of both forward and reverse primers, 1× Titanium Taq polymerase (Clontech, Mountain View, CA, USA) and 0.01 mg bovine serum albumin (BSA) were obtained with 1 µL of DNA template. FAD-linked oxidoreductase *easE*, a single copy ergot alkaloid synthesis gene, was amplified using designed primers: easE996f and easE1895r [22]. PCRs were run on a Mastercycler Pro S (Eppendorf, Mississauga, ON, Canada) using a touchdown protocol with initial denaturation at 95 °C for 3 min, followed by 5 cycles of 95 °C for 1 min, annealing at 63 °C (decrease of 1 °C per cycle) for 45 s and extension at 72 °C for 1 min 30 s, and then 30 cycles of 95 °C for 1 min, annealing at 58 °C for 45 s and extension at 72 °C for 1 min 30 s, with a final extension at 72 °C for 8 min. The PCR products were visualized on a 1% agarose gel with ethidium bromide treatment for 30 min at 200 V.

PCR products were amplified for Sanger sequencing using the ABI BigDye Terminator 3.1 cycling sequencing kit in a reaction volume of 10 µL, with BigDye Seq Mix diluted 1:8 with Seq buffer (Thermo Fisher Scientific, Ottawa, ON, Canada). The final volumes of each reagent were as follows: 1.75 µL of 5× sequencing buffer, 0.5 µL of BigDye Seq Mix and 0.5 µL of 3.2 µM primer, with reaction volumes increased to 9 μL using sterile high-performance liquid chromatography (HPLC) water. One microlitre of PCR product was added directly from the initial PCR amplification. The thermocycler profile used for the sequencing reactions had an initial denaturation step at 95 °C for 3 min, followed by 40 cycles at 95 °C for 30s, annealing at 58 °C for 15 s and extension at 60 °C for 3 min. An Applied Biosciences Prism 3130xl Genetic Analyzer (Life Technologies, Streetsville, ON, Canada) was used to generate DNA sequences and chromatograms. DNA sequences were edited and aligned to reference sequences in Geneious 11.1.5 to confirm identity. Once identity was confirmed, additional PCR and sequencing reactions were completed; DNA-directed RNA polymerase II subunit (*RPB2*) and translation elongation factor 1-α (*TEF1*-*α*) were amplified and sequenced using the above methods for phylogenetic construction.

### 5.3. Phylogenetic Analysis

A total of 43 reference sequences were downloaded and 2 outgroups were downloaded for each sequence, which included both the second largest subunit of the RNA polymerase II (*RPB2*) and elongation factor 1-α (*TEF1*-*α*) genes, and were aligned individually using MAFFT version 7 (online service https://mafft.cbrc.jp/alignment/server/ accessed on 22 January 2020). The Auto alignment strategy (FFT–NS–1, FFT–NS–2, FFT–NS–i or L–INS–i; depends on data size) was chosen. The alignments were visualized, verified and the two genes were concatenated using Geneious 11.1.5. Parsimony analysis was completed using PAUP* 4.0b10 [41]; heuristic searches with 50 replicates of random stepwise addition and tree bisection-reconnection branch swapping were conducted with a limit of 1,000,000 re-arrangements set for each replicate. Bootstrapping analyses were set with 100 replicates with full heuristic search of random stepwise addition of 50 replicates and a limit of 1,000,000 rearrangements per replicate.

### 5.4. HRMS Profiling of Sclerotium Extracts

Sclerotium fragments were individually frozen (−80 °C) and pulverized before being transferred into a 2 mL HPLC amber vial and extracted in 300 μL acetone:water (4:1, *v*:*v*) on a rotary shaker for 1 h (100 rpm). Extracts were centrifuged to pellet debris and 50 μL of the supernatant was transferred into a new HPLC vial for UPLC-HRMS profiling. Sclerotium extracts were profiled in a randomized injection order with solvent blanks interspersed between every six samples to assess for sample carryover and to aid in data curation during metabolomics processing. A reserpine standard was injected at the beginning of each sample sequence to confirm accurate calibration of the mass spectrometer and to aid in data alignment. Chemical profiling was completed using a Thermo Ultimate 3000 UPLC coupled to a Thermo LTQ Orbitrap XL HRMS and an UltiMate Corona VeoRS charged aerosol detector (Thermo Fisher Scientific Inc, Waltham, MS, USA). Chromatography was performed on a Phenomenex C_18_ Kinetex column (50 mm × 2.1 mm ID, 1.7 µm) with a flow rate of 0.35 mL/min, running a gradient of H_2_O (+0.1% formic acid) and ACN (+0.1% formic acid). The gradient started at 5% ACN, increased to 95% ACN over 4.5 min, was held at 95% ACN until 8.0 min, returned to 5% ACN by 9 min and was left to equilibrate for until 10 min before the next injection. The HRMS was operated in ESI^+^ mode (with a 100–2000 *m*/*z* range) using the following parameters: sheath gas (40), auxiliary gas (5), sweep gas (2), spray voltage (4.2 kV), capillary temperature (320 °C), capillary voltage (35 V) and tube lens (100 V). MS^n^ fragmentation was performed in high resolution on select ions in subsequent experiments using CID at 35 eV.

Additional experiments were performed to investigate the alkaloid diversity from LM04 and LM60 sclerotium extracts derived from rye. Samples were filtered through 0.2 μm PTFE membrane filters prior to analysis by nanoLC coupled to the Q-Exactive Plus mass spectrometer (Thermo Fisher Scientific). Chromatographic separation of metabolites was performed on a Proxeon EASY nLC II System (Thermo Fisher Scientific) equipped with a Thermo Scientific™ Acclaim™ PepMap™ RSLC C18 column (P/N ES800A), 15 cm × 75 μm ID, 3 μm, 100 Å, employing a H_2_O/ACN gradient (with 0.1% formic acid). Chromatography ran for 60 min at a flow rate of 0.25 μL/min: initiated with a linear gradient from 10 to 100% of ACN for 45 min, held at 100% ACN until 50 min, then decreased from 100 to 10% of ACN by 55 min and held at 10% until 60 min to equilibrate to starting conditions. The mass spectrometer used positive electrospray ionization (ESI) at an ion source temperature of 250 °C and an ionspray (Thermo Scientific™ EASY spray) voltage of 2.1 kV. The FTMS scan type was full MS/data-dependent (dd)-MS^2^. The parameters of the full mass scan were as follows: a resolution of 70,000, an auto gain control target under 3 × 10^6^, a maximum isolation time of 100 ms and an *m*/*z* range of 100–1500. The parameters of the dd-MS^2^ scan were as follows: a resolution of 17,500, an auto gain control target under 1 × 10^5^, a maximum isolation time of 100 ms, a loop count of top 10 peaks, an isolation window of *m*/*z* 2, a normalized collision energy of 35 and a dynamic exclusion duration of 10 s.

### 5.5. Metabolomics: Data Pre-Processing

UPLC–HRMS profiles of sclerotium extracts were compiled into a representative data matrix by converting metabolite mass spectral data into mass features (consisting of a retention time and a mass/charge ratio; RT and *m*/*z*), where each metabolite was represented by one or more mass features associated with various pseudomolecular ions such as protonated mass, salt adducts, neutral loses and charged fragments. UPLC-HRMS profiles of *C. purpurea* sclerotium extracts from the four grain crops were preprocessed together. Preprocessing of the acquired Xcalibur raw data files was carried out using MZmine 2.53 [42]. Masses were detected with a noise threshold of 1 × 10^4^. Chromatogram building was completed with the ADAP algorithm using a minimum group size of 6, a group intensity threshold of 1 × 10^6^ and a minimum highest intensity of 5 × 10^6^. RT and *m*/*z* tolerances were consistently set to 0.01 min and 0.005 *m*/*z* (or 5 ppm), respectively, throughout processing. Chromatogram deconvolution was carried out using the ADAP wavelets algorithm with a signal-to-noise threshold of 10, a minimum feature height of 2 × 10^6^, a coefficient/area threshold of 110, a peak duration rage of 0.00–0.50 and an RT wavelet range of 0.00–0.03. Isotopic peaks were then removed followed by alignment of peaks using the JOIN aligner method and a 20:10 ratio for *m*/*z* to RT weight. Peaks missing from the data matrix were then back filled using the same RT and *m*/*z* range gap-filling algorithm.

### 5.6. Metabolomics: Data Reduction

The resulting data matrix was exported from MZmine as a csv file and imported into R Studio where all mass features were normalized by mean. Mass features with retention times below 1 min and after 6 min were also removed due to the amplification of background noise at these time points. Prominent mass features occurring above cut-off thresholds in solvent blanks were subtracted from all feature values for a given mass feature to reduce false positives as well as to remove “system peaks” via in-house scripts. Correlation analysis was also preformed to group adducts and fragments. The applied correlation analysis used Pearson correlations (coefficient threshold set to 0.85) with an RT window of 0.02 to group mass features. Representative mass features for each of the groupings were selected based on evaluation of the aligned chromatograms for the most consistent peak shape, relative intensity and preference for the most likely [M + H]^+^ ion by manual determination.

### 5.7. Metabolomics: Binary Matrix Data Transformation

Mass feature peak area values were converted to binary presence/absence matrices (to facilitate comparison between the various host crops) using a peak area threshold value of 5 (where values lower then 5 were denoted as 0 and those greater than denoted as 1). For each individual host, mass features of host crop sclerotium replicates were individually averaged to generate a single representative value for each isolate. Mass features with averaged values greater then 0.65 were denoted as 1, while values lower were set to 0, to ensure that only mass features consistently produced by isolates were included in the resulting host sclerotia binary matrix (4/6 replicates). An untargeted consensus binary heatmap representing mass feature frequency of occurrence in sclerotia produced on the four host crops was constructed to further explore the overall mass feature diversity amongst the strains and between classes. The mass features across the four host binary matrices were then summed for each isolate to create a consensus frequency phenotype matrix consisting of frequency values ranging from 0 to 4, with 0 indicating the feature as not detected for the isolate on any of the examined hosts and 4 indicating the presence in sclerotia derived from all hosts.

### 5.8. Metabolomics: Multivariate and Univariate Analyses

Multivariate and univariate analyses was completed using the “MUMA” R package with applied Pareto scaling and half minimum imputation on zero values. An initial unsupervised Euclidean Ward hierarchical clustering was performed individually for each host crop to reveal the underlying structure within the data. Metabolomic phenotype heatmaps were generated using the binary data matrices in the “pheatmap” R package, with row and column dendrograms calculated using “ward.D2” clustering of the Euclidean distance matrix. A second metabolomic phenotype heatmap was also generated using only mass feature associated with ergot alkaloid biosynthesis to further demonstrate the dichotomy in ergot alkaloid production across the two classes. Discriminant multivariate analyses (random forest and OPLS-DA) were performed using the original non-binary data matrix to determine which mass features contributed most to class separation for each crop host. The supervised random forest analysis was completed using the “randomForest” R packages with the identified classes used for trained data and the OPLS-DA was executed using the “MUMA” R package.

In the random forest analysis, two indices (mean decrease accuracy and mean decrease Gini) are used to assess variable importance associated with class separation. Mean decrease accuracy uses permuting “out of bag” samples to compute the importance of each variable in the predictive accuracy of the random forest [43]. Mean decrease Gini is a measure of how important a variable is in contributing to node and leaf homogeneity across all of the decision trees, where the larger the Gini index value, the greater the importance the variable has in terms of classification in the random forest model [43]. Both mean decrease accuracy and mean decrease Gini were used to rank the influence of the mass features on class separation between sclerotium extracts based on variances in relative peak intensity patterns.

### 5.9. Metabolomics: Mass Feature Annotation

Mass feature annotations were completed using an in-house *C. purpurea* reference database created from literature reports [33,44]. Putative ergot alkaloid annotations were assigned based on the mass spectral accuracy (±5 ppm) and relative elution order. MS^n^ experiments were used to confirm ergot alkaloid annotations based on common fragmentation patterns expected from reports in the literature. MassWorks^TM^ software (v5.0.0, Cerno Bioscience, Las Vegas, NV, USA) was used to improve spectral accuracy and confirm the molecular formulas of annotated ions. The sCLIPS searches were performed in dynamic analysis mode with allowances for the elements C, H, N and O set at a minimum of 1 and a maximum of 100. Charge was specified as 1, mass tolerance was set to 5 ppm and the profile mass range was −1.00 to 3.50 Da.

To support of ergot alkaloid annotations, the obtained nanoLC-HRMS/MS data of rye sclerotia extracts from LM04 and LM60 were compared to the ESI+ LCMS spectral database downloaded from the MassBank of north America website in August 2021. MS^2^ spectral comparisons were made using MSDIAL [45] and again using default parameters in the GNPS web-based workflow for molecular networking [46]. Top database matches supported annotations for ergometrine, methylergometrine, ergovaline, ergosine, ergotamine, ergocornine, ergocryptine, ergocristam and ergocristine. Most ergot alkaloid peptam/peptide precursors are not represented in spectral databases at this time. Additionally, MS/MS spectral matching is unable to reliably differentiate between very similar spectra from the same parent ion *m*/*z*, such as the ‘-ine’ and ‘-inine’ form of ergot alkaloids.

### 5.10. Genomics: DNA Isolation

Isolates were cultured on PDA and harvested and ground using liquid N_2_. Genomic DNA was extracted using a cetyltrimethyl ammonium bromide (CTAB) protocol [47,48]. DNA samples were assessed by running on a 1% agarose gel for the presence of DNA shearing and RNA, while DNA integrity was evaluated using TapeStation and Genomic DNA ScreenTapes (Agilent, Santa Clara, CA, USA) and impurities were evaluated using DropSense 16 (Trinean, Pleasantan, CA, USA). gDNA was quantified using a Qubit^®^ 2.0 Fluorometer (Invitrogen by Life Technologies, Carlsbad, CA, USA) before submission for in-house sequencing (Molecular Technologies Laboratory, Ottawa Research & Development Centre, Agriculture and Agri-Food Canada) where DNA libraries were prepared and loaded onto a FLO-MIN106 flow cell and run with a MinION (Oxford Nanopore Technologies, Oxford, UK) for 48 h.

### 5.11. Genomics: Genome Assembly and Ergot Alkaloid BiosynthetiFc Gene Cluster Annotation

Genome assembly of LM72, LM04 and LM60 was performed with CANU v1.8 [49] using the sequenced Nanopore reads with default settings and an estimated genome size of 35 Mb. Two rounds of correction were applied to the resulting assemblies: the first round was performed using Nanopolish v. 0.13.2 [50], and the second round used Pilon v1.23 [51], which corrected the nanopolished CANU assemblies using Illumina reads that were mapped with BWA v0.7.17 [52]. Assemblies were annotated using the Funannotate v1.8.8 pipeline [53] using default settings, with predicted proteins from *C. purpurea* isolate 20.1 supplied as protein evidence to assist gene modeling. Libraries of repetitive elements were generated using RepeatModeler2 [54], and were identified, where possible, using the 2018 Repbase database of annotated transposons [55]. Repetitive elements in intergenic regions between *lpsA1* and *lpsA2* were annotated by searching for blast hits in the RepeatModeler2-generated repetitive element libraries. Illumina-based genome assemblies for LM04, LM30, LM60, LM207, LM232, LM233, LM464 and LM479 were previously published and are publicly available in the NCBI database [30].

## Figures and Tables

**Figure 1 toxins-13-00861-f001:**
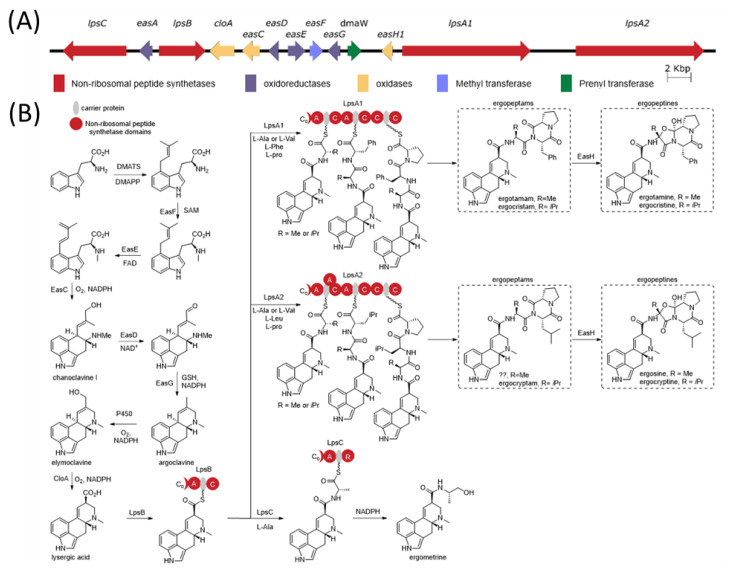
Lysergic acid and ergot alkaloid biosynthesis. (**A**) The organization of the ergot alkaloid biosynthetic gene cluster from *C. purpurea* 20.1. (**B**) The biosynthetic pathway for lysergic acid and the ergot alkaloids (C = condensation; A = adenylation; R = reductase domain; C_0_ = C-terminal portion of a C domain).

**Figure 2 toxins-13-00861-f002:**
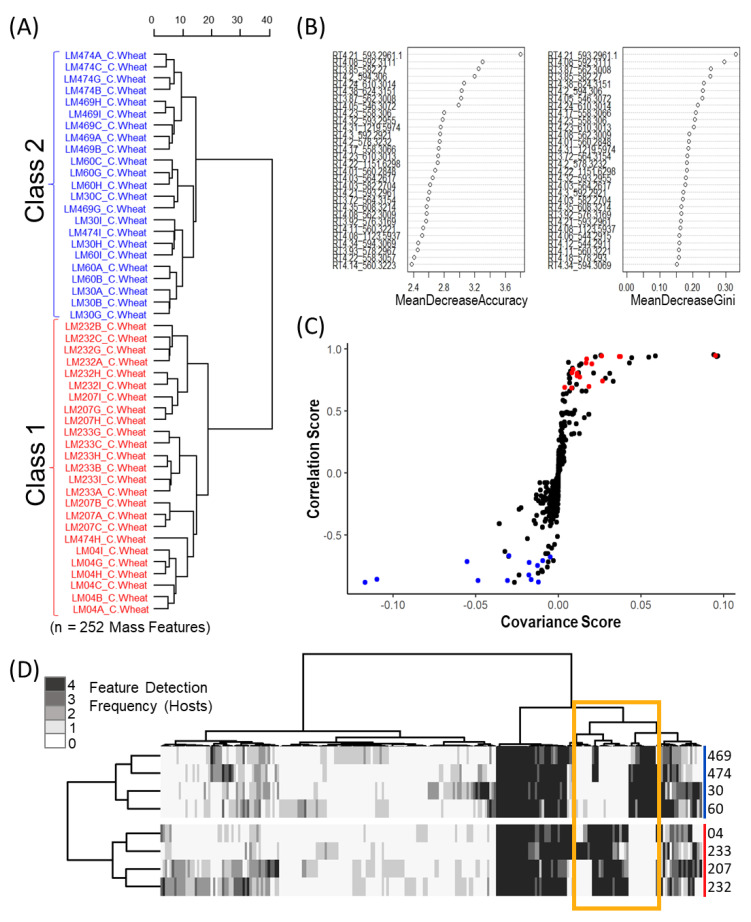
Untargeted analysis support two putative classes of *C. purpurea* strains grown on ‘common’ wheat. (**A**) Euclidean Ward hierarchical clustering of pseudo-binary (presence/absence) mass feature data of replicate sclerotia. Labels in Red and Blue represent specimens identified in Class 1 and Class 2, respectively, with (n = 6) replicates per specimen. (**B**) Top 30 mass features associated with class formation as determined via random forest analysis of raw data (left: mean decrease in accuracy; right: mean decrease in Gini of raw data of replicate sclerotia). (**C**) S-plot from OPLS-DA highlighting top 30 features identified by random forest analysis of replicate sclerotia, features in red and blue represent specimens identified in Class 1 and Class 2. (**D**) Consensus phenotype heatmap constructed using Ward D2 hierarchical clustering of Euclidean distances between all conserved mass features across the 4 hosts (n = 518 mass features). The mass features indicated in yellow are the likely drivers of this dichotomy.

**Figure 3 toxins-13-00861-f003:**
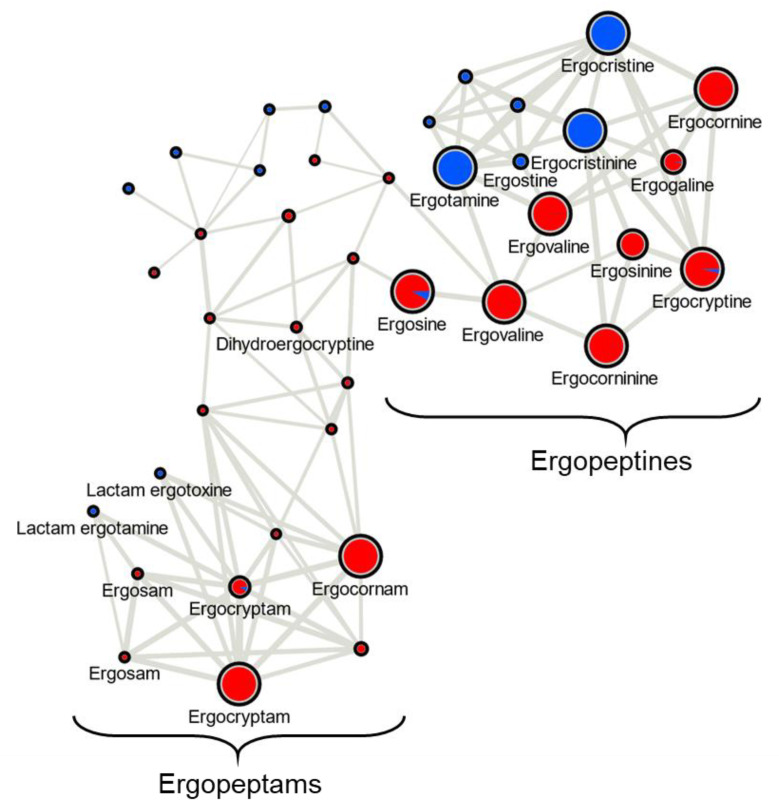
Molecular networking map of ergot alkaloids detected in sclerotium extracts from LM04 (Class 1—red) and LM60 (Class 2—blue) and grown on rye. The sizes of the pie charts reflect the intensity of the precursor ions (summed) and pie chart ratio is calculated from the *m*/*z* peak areas. Lines connect nodes if cosine scores comparing MS^2^ scans are above 0.7, with thickness increasing as the cosine score increases. Some annotations are duplicated where the MS^2^ analysis was unable to differentiate putative stereoisomers.

**Figure 4 toxins-13-00861-f004:**
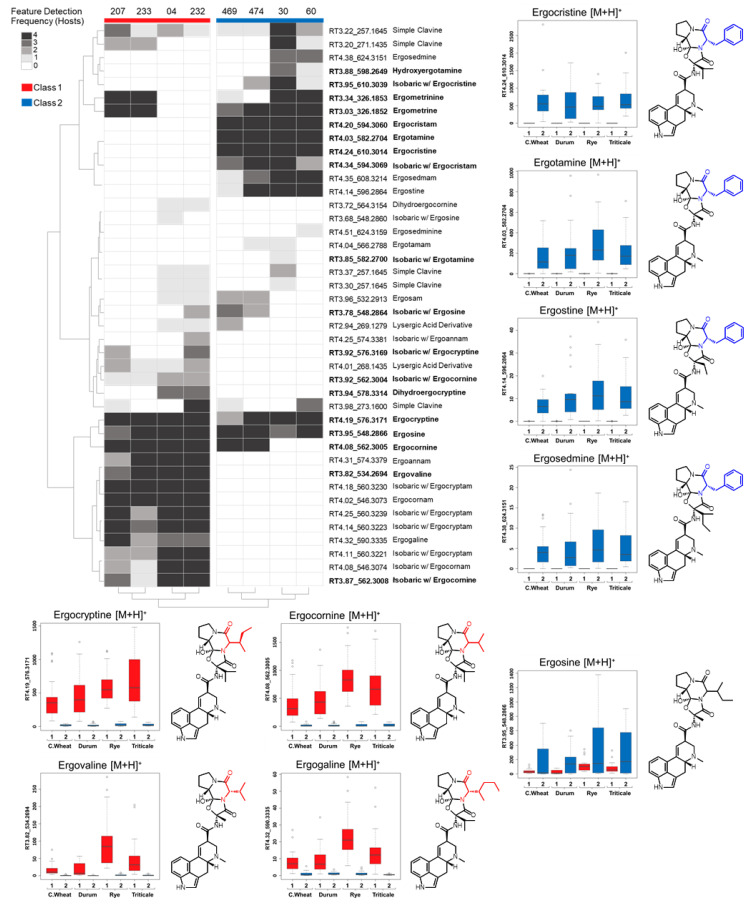
Consensus phenotype heatmap constructed using hierarchical clustering of representative mass features of ergot alkaloids and related derivatives demonstrating the frequency of occurrence and dichotomy in *C. purpurea* classes. Boxplots comparing frequency of occurrence by host crop of mass features accompanied with associated ergopeptine. In the boxplots, outliers are represented by black ovals; key structural differences in the ergopeptine aa2 residue are coloured by class association (blue = Class 1, red = Class 2).

**Figure 5 toxins-13-00861-f005:**
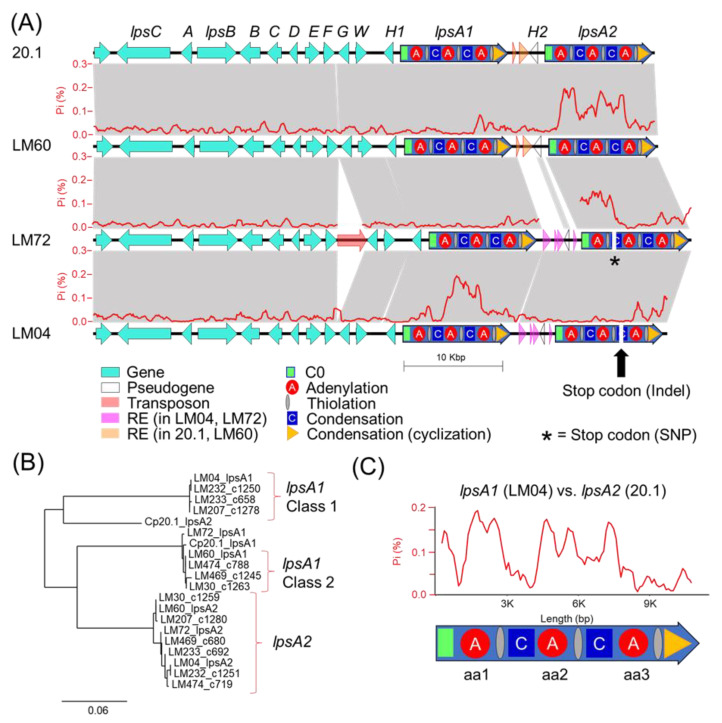
(**A**) Synteny analysis between ergot alkaloid biosynthetic gene clusters in *C. purpurea* strains 20.1, LM60, LM72 and LM04. Dark blue genes represent *lpsA* NRPSs, all other genes are labelled in light blue (see Figure 1 for gene classifications). Gene letters have the prefix ‘*eas*’ removed for brevity, with the exception of *B*, which represents *cloA*, and *W*, which represents *dmaW*. Transparent arrows represent repetitive elements. Grey blocks represent syntenic DNA above 88% nucleotide identity, with the exception of the block connecting *easH2* pseudogenes located between LM60 and LM72, which have only 50% nucleotide identity. The red line graph overlaid onto the syntenic blocks represents a running average of the proportion of polymorphic DNA sites as calculated in 100 bp stepwise increments of a 500 bp sliding window, in each instance comparing the sequences above and below, and should be interpreted as aligning to the track below. NRPS domains are overlaid on *lpsA* genes that were predicted by antiSMASH 6.0. A yellow diamond is placed where a frameshift mutation was predicted to have disrupted the second condensation domain of lpsA2 in LM04 via premature stop codon insertion. (**B**) A phylogenetic comparison of the nucleotide sequence encoding the second A-domain (aa2) of all *lpsA* genes from all strains included in this study. The tree shows the clustering of all aa2 A-domains from predicted *lpsA1* genes correlates to the metabolomic clustering of strains identified as ‘Class 1’ vs. ‘Class 2’. (**C**) Comparison of polymorphic site frequencies between *lpsA2* of *C. purpurea* strain 20.1 and *lpsA1* of LM04.

## Data Availability

Annotated ergot alkaloid biosynthetic gene cluster sequences for LM04, LM60 and LM72 were uploaded to the NCBI nucleotide database under the following accession numbers respectively: OK662595, OL348384, and OL348385.

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
