# Peer review of "Evolution of the Ergot Alkaloid Biosynthetic Gene Cluster Results in Divergent Mycotoxin Profiles in Claviceps purpurea Sclerotia"

_toxins, 2021, doi:10.3390/toxins13120861_

Round 1
Reviewer 1 Report
The authors used an untargeted metabolomics to investigate Ergot Alkaloid profile of C. purpurea strains. This is an important and timely research, which further adds to current understanding in terms of Clavicep infection and contamination of food and feed by EA. The manuscript is well written; however, the discussion section is scanty or too restricted. Addition of more paragraphs on:
- Occurrence of prominent EAs in food and feed
- Toxic effects of EAs on human and animal health
- Impact of climate change in relation to evolution of EA production by C. purpurea.
would improve the quality of this manuscript.
Author Response
In response to Reviewer 1 comments:
The discussion section has now been expanded to include two additional paragraphs. One paragraph was added to include previous accounts in the published literature of the occurrence of prominent ergot alkaloids in Claviceps sclerotia - to build upon the discussion of observed differences in ergot alkaloid expression patterns. A second paragraph was added to describe the toxic effects of ergot alkaloids on human and animal health – highlighting the complexities of ergotism in terms of acute exposure and subtle, prolonged exposure and the associated impacts on livestock.
Regarding the suggested addition of discussion around the impact of climate change in relation to the evolution of EA production, we acknowledge that climate change will likely result in increased incidents of ergot and potentially exacerbate the issues currently posed by C. purpurea in agriculture. However, we feel that predictions as to how climate change will influence the evolution of EA production – in the context of the dichotomy in EA production observed in this manuscript – are a little premature and we feel that more evidence must first be collected before speculating on exactly how these environmental conditions will drive evolution of the gene cluster.
Reviewer 2 Report
The authors addressed a relevant problem, especially for Canadian crops and feed safety. The combination of molecular and mass spectrometry approaches uncover the variation in alkaloid content across Canadian C. purpurea isolates. The work is well written. The methodology was designed according to the objectives. The conclusions are based on the main findings.
- The discussion should be enriched with a comparison taking account results previously published in the literature.
- A table summarizing the major specialized (secondary) metabolites identified, with their respective mass and retention time should be helpful.
Author Response
In response to Reviewer 2 comments:
The discussion section has now been expanded by two paragraphs, one of which addresses comment 1 from Reviewer 2. The discussion was enriched with a paragraph outlining historical results from the published literature surrounding ergot alkaloid production in Claviceps spp. Focus was placed upon observed differences in ergot alkaloid content between Claviceps strains – and how this differences related to the dichotomy in ergot alkaloid production observed in the research study. A second paragraph was added to describe the toxic effects of ergot alkaloids on human and animal health – highlighting the complexities of ergotism in terms of acute exposure and subtle, prolonged exposure and the associated impacts on livestock.
With respect to comment 2 - as suggested, a table summarizing the ergot alkaloid mass features used for Class 1 / Class 2 separation has been added to the supplementary information and referred to in the manuscript (Table S-2). This table includes retention times, observed m/z of the protonated ion, theoretical m/z, delta ppm, and molecular formula.
Reviewer 3 Report
The paper “Evolution of the ergot alkaloid biosynthetic gene cluster results in divergent mycotoxin profiles in Claviceps purpurea sclerotia” provide useful information about the variation in alkaloid content across Canadian C. purpurea isolates. The study was well conducted; the experimental design and the statistical analysis have been adequately thought out and carried out. The manuscript is easy to read; the methodology and the results have been well described.
At current state, the manuscript is suitable for publication after minor revisions.
Minor comments/suggestions follow.
Line 132: which investigation? Please, explain better
Line 202: Please, change Figure 4 to Figure 5
Supplementary materials: There is no trace of Figures S-2, S-3 and S-4 in the paper. Please, insert and explain them in the text.
Author Response
In response to Reviewer 3 comments:
Line 132: which investigation? Please, explain better
(The investigation referred to all of the work presented in Section 2.2 of the manuscript – that follows in the subsequent section after the sentence. To avoid confusion, a note has been place in brackets after the text “Further investigation…”, referring the reader to section 2.2 of the manuscript)
Line 202: Please, change Figure 4 to Figure 5
(This change has been made)
Supplementary materials: There is no trace of Figures S-2, S-3 and S-4 in the paper. Please, insert and explain them in the text.
(reference to Figures S-2, S-3, and S-4 are present in the manuscript in several places (lines 107, 112, 114), and can be found adjacent to the reference for Figure 2 of the revised manuscript)